# A Denture Use Model Associated with *Candida* spp. in Immunocompetent Male and Female Rats

**DOI:** 10.3390/jof8050466

**Published:** 2022-04-30

**Authors:** Vinicius Tatsuyuji Sakima, Yuliana Vega-Chacón, Paulo Sergio Cerri, Bhumika Shokeen, Renate Lux, Ewerton Garcia de Oliveira Mima

**Affiliations:** 1Laboratory of Applied Microbiology, Department of Dental Materials and Prosthodontics, School of Dentistry, Araraquara, São Paulo State University (UNESP), São Paulo 14801-903, Brazil; sakima888@gmail.com (V.T.S.); yuliana.v.chacon@unesp.br (Y.V.-C.); 2Laboratory of Histology and Embryology, Department of Morphology, Genetics, Orthodontics and Pediatric Dentistry, School of Dentistry, Araraquara, São Paulo State University (UNESP), São Paulo 14801-903, Brazil; paulo.cerri@unesp.br; 3School of Dentistry, University of California at Los Angeles (UCLA), Los Angeles, CA 90095, USA; bhumikas@ucla.edu (B.S.); rlux@dentistry.ucla.edu (R.L.)

**Keywords:** rats, dentures, oral candidiasis, *Candida albicans*, *Candida glabrata*, *Candida tropicalis*

## Abstract

Denture stomatitis (DS) is a common infection in denture wearers, especially women. This study evaluated the induction of DS using acrylic devices attached to the palate of rats combined with inoculation of *Candida* spp. Immunocompetent male and female rats received a carbohydrate-rich diet. Impressions were taken from the rats’ palate to individually fabricate acrylic devices. Mono- and multispecies biofilms of *C. albicans*, *C. glabrata*, and *C. tropicalis* were grown on the devices, which were then cemented on posterior teeth and kept in the rats’ palate for four weeks. Microbial samples from the palate and the device were quantified. Oral microbiome of rats inoculated with *C. albicans* was analyzed by 16S rRNA gene sequencing. Log_10_(CFU/mL) were analyzed by mixed or two-way MANOVA (*α* = 0.05). *Candida* spp. and acrylic device did not induce palatal inflammation macroscopically nor microscopically. Although there was an increase (*p* < 0.001) of the total microbiota and female rats demonstrated higher (*p* = 0.007) recovery of *Candida* spp. from the palate, the gender differences were not biologically relevant. The microbiome results indicate an increase in inflammatory microbiota and reduction in health-associated micro-organisms. Although *Candida* spp. and acrylic device did not induce DS in immunocompetent rats, the shift in microbiota may precede manifestation of inflammation.

## 1. Introduction

Denture stomatitis (DS) is considered the most common oral lesion among denture users [1], with a higher prevalence in women [2,3]. Although the etiology of DS is multifactorial, infection by *Candida* spp., especially *Candida albicans*, is considered the main etiological factor [4]. Oral micro-organisms have a high affinity for the denture acrylic, to which they adhere and develop biofilms [5,6]. Among the species of *Candida* isolated from DS, *C. albicans* is the most prevalent species [4], which is a commensal micro-organism in different body sites in most healthy individuals. However, *C. albicans* thrives as a pathogen under certain host conditions, such as immunosuppression, xerostomia, broad-spectrum antibiotic use, and bacterial dysbiosis [7,8]. In addition to *C. albicans*, other *Candida* species such as *C. glabrata* and *C. tropicalis* are also present in patients with DS and are commonly co-isolated with *C. albicans* [9,10,11,12,13]. These species have also been isolated in cases of candidemia [14,15,16] and have shown resistance or reduced susceptibility to antifungal agents [14,17]. Although these species are clinically found together with *C. albicans* in mixed biofilms, the dynamics among the different *Candida* species and their potentially synergistic virulence has not been widely studied [18,19].

Animal models have been used to investigate the pathogenesis of oral candidiasis and therapeutic approaches [7,20,21]. Many studies have used a murine model of oral candidiasis, in which immunosuppressed female mice are inoculated with *C. albicans*, resulting in white lesions on the tongue [22,23,24]. However, the use of immunosuppressed mice reproduce neither the erythematous characteristic of the palate found clinically in DS, nor the systemic condition of immunocompetent denture users. For this reason, some studies have fitted acrylic devices on the rats’ palate to induce DS [25,26,27,28,29,30,31,32]. These studies used male rats in contrast to the murine model that employed female mice. As DS is more prevalent in females [2,3], the comparison of genders in animal models is important for its reproducibility and relevance to the clinical conditions.

Regarding the pathogenesis of non-*albicans Candida* species in rat models of DS, a previous study observed that *C. glabrata* neither resulted in inflammation of the rats’ palate nor exacerbated the infection caused by *C. albicans* [33]. Similarly, another study verified that *C. glabrata* and *C. tropicalis* alone did not induce inflammation in the palatal mucosa of rats, while only one serotype of *C. albicans* induced it [34]. In a murine model of oropharyngeal candidiasis, no infection on the mice’s tongue was observed when *C. glabrata* alone was orally inoculated. However oral colonization by *C. glabrata* was possible after previous infection or co-inoculation with *C. albicans* [35]. Another study observed that only one of three clinical isolates of *C. tropicalis* produced infection on the tongue of female rats [36]. Different from the non-filamentous *C. glabrata*, *C. tropicalis* produces true hyphae and is considered the most virulent non-*albicans Candida* species [37]. However, to the best of our knowledge, co-infection of *C. albicans* and *C. tropicalis* has not been tested in rodent models of oral candidiasis. Thus, this study evaluated the induction of DS by *C. albicans*, *C. glabrata*, and *C. tropicalis* in a rat model using fitted acrylic devices.

## 2. Materials and Methods

### 2.1. Animals

Forty-seven male and 44 female albino Wistar rats (*Rattus norvegicus*) weighing 400–450 g and 250–300 g, respectively were used in this study (provided by São Paulo State University, UNESP—Botucatu). The research protocol was approved by the Ethical Committee for Animal Use (process 25/2018) by the School of Dentistry, Araraquara, UNESP. Rats received water *ad libitum* and standard chow diet and were kept individually in a vivarium with a 12/12 h light/dark cycle and temperature between 22 and 24 °C. Experiments were performed with 90-day-old rats. For intraoral procedures, animals were anesthetized by intramuscular injection with ketamine at 75 mg/kg (Dopalen^®^, Ceva, Sao Paulo, Brazil) and xylazine at 8 mg/kg (Anasedan^®^, Ceva, Sao Paulo, Brazil). Baseline sample collection was performed by scrubbing a sterile swab in the oral cavity of each animal. The swab was immersed in 1 mL phosphate-buffered saline (PBS—0.136 M NaCl, 1 mM KH_2_PO_4_, 2 mM KCl, 10 mM Na_2_HPO_4_, pH 7.4), vortexed, serial-diluted and plated on Brain-Heart Infusion (BHI, Difco Detroit, MI, USA) and Sabouraud Dextrose Agar (SDA; Acumedia Manufacturers Inc., Baltimore, MD, USA) plates, which were incubated at 37 °C for 48 h to assess total microbiota and fungal growth (CFU/mL), respectively. As an inclusion criterion, only animals negative for indigenous candidal growth were selected. Animals also received a carbohydrate-rich diet (40% sugar, 30% starch, and 30% corn flour) during the whole experimental period (prior to the installation of the intraoral acrylic device until the euthanasia) to maintain *Candida* spp. in the oral cavity without immunosuppression. Rats were monitored weekly for any sign of stress and weight loss.

### 2.2. Custom-Fitted Intraoral Acrylic Device

Intraoral devices retained by cementation in the maxillary molars were fabricated covering the posterior teeth and the hard palate. First, individual acrylic trays were made from a maxillary impression of a 90-day-old rat dry skull. With the individual trays, impressions of the palate were taken from each animal under anesthesia, using polyether (Impregum, 3M, Sao Paulo, Brazil) impression material. The impressions were poured with type IV dental stone (Herodent, Vigodent S.A., Rio De Janeiro, Brazil) to obtain the casts. Individual waxing of the palatal region covering the molar teeth was performed on the casts. Then, the devices were produced using thermo-polymerizing acrylic resin (Clássico^®^, São Paulo, Brazil) replacing the wax from a flasking procedure. Relief was performed on the molar teeth area of the device to be cemented with self-curing acrylic resin (Clássico^®^, São Paulo, Brazil). A perforation with a diameter of a 200-μL pipette tip was made in the center of the device to enable intermittent fungal inoculations without removing the device [38]. The devices were maintained in distilled water for two days. Afterwards, they were individually transferred to 200 mL of sterile distilled water and sterilized by microwave irradiation at 650 Watts for 3 min [39] before the biofilm formation.

### 2.3. Candida *spp.* Growth, Biofilm Formation on Devices, Oral Inoculations and Experimental Groups

For biofilm formation and oral inoculations, a laboratory strain of *C. albicans* (Ca, SC5314) and standard strains of *C. glabrata* (Cg, ATCC 2001) and *C. tropicalis* (Ct, ATCC 4563) were thawed and grown in SDA for 48 h. Colonies of each strain were transferred to Yeast Nitrogen Broth (YNB, Difco, InterLab, Detroit, MI, USA) medium, which was incubated at 37 °C for 16 h. The fungal suspensions were diluted 1:10 in fresh YNB and incubated at 37 °C until their optical density at 540 nm (OD_540_) reached the mid-log phase of growth (mean ± standard deviation, SD) [Ca: 0.536 ± 0.062 arbitrary units (a.u.); Cg: 0.756 ± 0.038 a.u.; Ct: 0.429 ± 0.056 a.u.]. Thus, fungal suspensions were standardized at concentrations of 1.38 × 10^7^ ± 4.95 × 10^6^, 1.69 × 10^7^ ± 2.72 × 10^6^, and 3.55 × 10^6^ ± 1.33 × 10^5^ colony forming units per milliliter (CFU/mL) for Ca, Cg, and Ct, respectively [or 7.11 ± 0.16, 7.22 ± 0.07, and 6.53 ± 0.02 log_10_(CFU/mL), respectively]. When two or three species were combined for oral inoculation or biofilm formation, the volume of each fungal suspension was proportioned according to its standard concentration and then mixed.

These standardized fungal suspensions were individually used for biofilm formation and oral inoculation. For biofilm formation, the devices were transferred individually into the wells of a 24-well plate and incubated with 2 mL of fungal suspension containing single or mixed species for 90 min at 37 °C. Then, the devices were washed twice with PBS and incubated with 2 mL of Roswell Park Memorial Institute (RPMI) 1640 medium—bicarbonate free (Sigma-Aldrich, St. Louis, MO, USA), buffered with morpholinepropanesulfonic acid (MOPS; Sigma-Aldrich), and supplemented with 2% D-glucose (Synth, São Paulo, Brazil), pH 7.0 (RPMId) for 48 h at 37 °C. After biofilm formation, the devices were washed twice with PBS before being attached on the animal’s palate. For oral inoculation, 10 mL of single or mixed-species fungal suspension was centrifuged at 6000× *g* for 10 min at 4 °C, washed twice with PBS and the entire pellet was pipetted on the animal’s palate (one pellet per animal). Immediately after oral inoculation, the device was cemented on the molar maxillary tooth using a self-curing acrylic resin (Clássico, São Paulo, Brazil). Additional oral inoculations were performed twice in an interval of three days, and the fungal pellet was introduced into the cavity of the palate through the orifice of the device cemented inside the oral cavity. The devices were kept inside the rats’ oral cavity for four weeks [33]. The animals of each sex were randomly divided into the following study groups (*n* = 5 each, except negative control and device control groups with *n* = 2 and 3, respectively): Negative Control (NC): without intraoral device and without fungal inoculation; Device Control (DC): with intraoral device and without fungal inoculation; Group *C. albicans* (Ca): with intraoral device and with inoculation of Ca; Group *C. glabrata* (Cg): with intraoral device and with inoculation of Cg; Group *C. tropicalis* (Ct): with intraoral device and with inoculation of Ct; Group *C. albicans* and *C. glabrata* (CaCg): with intraoral device and with inoculation of Ca and Cg; Group *C. albicans* and *C. tropicalis* (CaCt): with intraoral device and with inoculation of Ca and Ct; Group *C. glabrata* and *C. tropicalis* (CgCt): with intraoral device and with inoculation of Cg and Ct; and finally Group *C. albicans*, *C. glabrata* and *C. tropicalis* (CaCgCt): with intraoral device and with inoculation of Ca, Cg and Ct.

### 2.4. CFU Analysis

After four weeks, the devices were carefully removed from the mouth of the anesthetized animals. The devices were further immersed in tubes containing 2 mL of sterile PBS. A final sample collection from the palate was also performed using a sterile swab, which was rubbed for 1 min on the palate and immersed in 1 mL of PBS. Devices and swabs were vortexed for 2 min, subjected to tenfold serial dilutions, which were plated (25 μL) in duplicate on BHI, CHROMAgar *Candida* (CHRO, Difco Detroit, MI, USA) and SDA medium plates, and incubated at 37 °C for 48 h. The colonies of *Candida* spp. in CHROMAgar *Candida* were presumptively identified (green: Ca; pink: Cg; blue: Ct) and counted to determine the CFU/mL values for each species. Colonies on BHI (total microbiota) and SDA (fungi) were also quantified (CFU/mL).

### 2.5. Macroscopic Analysis

Photographs of the palate were taken (Canon T6i, Tokyo, Japan) before the devices were cemented in the animals’ oral cavity and also at the end of the experiment after removal of the acrylic devices to evaluate the mucosal inflammation. Bleeding in the gingival sulcus was scored as 0 (no bleeding), 1 (mild bleeding: blood observed only in the marginal gingival), 2 (moderate bleeding: blood covering teeth), and 3 (intense bleeding: blood covering tooth and mucosae).

### 2.6. Euthanasia and Blood Cell Count Analysis

After the devices were removed from the oral cavity of the anesthetized rats, blood of each animal was collected via renal artery for blood cell count (Laboratório São Lucas, Araraquara, Brazil), and the mucosa of the hard palate was surgically removed for histological analysis. Thereafter, animals were euthanized with a lethal dose of ketamine hydrochloride by intramuscular injection into the femur.

### 2.7. Histopathological Analysis

Following fixation in 4% formaldehyde buffered at pH 7.2 with 0.1 M sodium phosphate, the mucosal samples were processed for paraffin embedding. The sections (5-μm thick) were adhered to glass slides and submitted to the periodic acid–Schiff (PAS) histochemical method and counterstained with hematoxylin for observation using a light microscope (BX-51, Olympus Corporation, Tokyo, Japan). The morphological changes were evaluated according to the presence of yeast and filamentous cells, the organization of the epithelial layer, and the presence of inflammatory cells in the connective tissue.

### 2.8. DNA Extraction and 16S rRNA Gene Sequencing

The DNA from samples was extracted using the MasterPure DNA purification kit (Epicenter Technologies, Madison, WI, USA) according to the manufacturer. DNA quantity, purity, and integrity were evaluated by optical density at 260 nm (Nanodrop, DeNovix DS11, Wilmington, NC, USA), ratio 260/280, and electrophoresis on 0.8% agarose gel, respectively. The DNA was extracted only from the Ca group swabs collected both at the baseline (before cementing the device) and after the four-week period of device use to compare the possible shift in the oral microbiome.

The 16S ribosomal RNA gene was amplified according to the HOMINGS protocol (homings.forsyth.org, accessed on 10 March 2022) using V1–V3 primers (F:5′**TCGTCGGCAGCGTCAGATGTGTATAAGAGACAG**AGAGTTTGATCMTGGCTCAG3′,R:5′**GTCTCGTGGGCTCGGAGATGTGTATAAGAGACAG**ATTACCGCGGCTGCTGG **3**) instead of the V3–V4 primers of the original protocol. Briefly, the V1–V3 region of the 16S rRNA gene was amplified using gene specific primers with Illumina overhang adapters. The polymerase chain reaction (PCR) mixture included 25 ng DNA, 0.4 μM each of the forward and reverse primers, and 1× of the HF Taq polymerase Master mix (NEB) in a total volume of 25 μL. The PCR cycles consisted of an initial denaturation step of 98 °C for 3 min followed by 35 cycles of 98 °C for 30 s, 55 °C for 30 s, 72 °C for 30 s with a final extension of 72 °C for 5 min. Three PCR reactions were carried out for each sample and pooled together to reduce bias prior to cleaning with AMPure XP beads (Beckman Coulter, A63881). The amplicons were indexed using the Nextera XT Index kit (Illumina) and purified again with AMPure XP beads (Beckman Coulter, A63881). After quantification with the DNA KAPPA kit (ROCHE Diagnostics), equal amounts of each sample were pooled into a single library. The quality and quantity of the library were checked at Technology Center of Genetics and Bioinformatics Core at UCLA before Miseq (2 × 300 bp) paired end sequencing on the Illumina platform.

### 2.9. Sequencing Data Analysis

Demultiplexed single-end sequences were obtained from Technology Center of Genetics and Bioinformatics Core at UCLA and imported into QIIME 2 (v2020.11) [40]. Low quality sequences containing bases with Phred quality values < 20 were trimmed and denoised using the DADA2 package [41]. The amplicon sequence variants (ASVs) generated with forward reads were taxonomically assigned by comparison to the HOMD database [42]. Alpha and beta diversity analyses were performed using the core metrics plugin in QIIME 2. The Shannon’s index diversity measure was used for calculating alpha diversity, while weighted unifrac was used for assessing beta diversity.

### 2.10. Statistical Analysis

Statistical analysis was performed for CFUs recovered from palate/device and the body mass (g) of animals. Normality, homoscedasticity and sphericity were evaluated by Shapiro–Wilk, Levene, and Mauchly’s tests, respectively. Body mass values were analyzed by repeated measures ANOVA and log_10_(CFU/mL) values were submitted to mixed or two-way MANOVA (sex and group) and post-hoc Tukey tests. The significance level adopted was 5%. All analyses were performed using SPSS software version 20.0 (SPSS Inc., Chicago, IL, USA). The photographs, histopathological and SEM images were analyzed descriptively.

Statistical comparisons for alpha diversity and relative abundance were completed using GraphPad Prism (version 9.1.0, GraphPad Software, San Diego, CA, USA). The beta diversity measures were evaluated using analysis of similarity (ANOSIM) with 999 permutations in QIIME 2 [40].

## 3. Results

### 3.1. Animals’ Body Mass

After cementing the intraoral devices in the animals’ oral cavity, the rats lost a significant amount of body mass (g) on the 10th and 13th day (Appendix A). After this period, the animals showed a gradual increase in their body masses throughout the evaluation period. During the last two weeks with the device, the rats’ body masses were significantly higher (*p* < 0.001) than those measured at baseline. This suggests that the animals were able to feed with the intra-oral device. Crumbs from the solid pellet chow were also observed below the individual cages, which indicated that the animals were able to gnaw the chow. During the entire experimental period, 11 animals died (7 males and 4 females) and were replaced by other animals to keep the same number of animals (*n* = 5 for each gender) in each group.

### 3.2. Recovery of Micro-Organisms

The recovery of the total cultivable microbiota (BHI agar) from the rats’ palate demonstrated a significant increase (*p* < 0.001) of the log_10_(CFU/mL) values after the four-week period with the device, compared with the initial oral recovery (before cementing the device), in all the groups and both female and male rats. The lowest increase was observed for the Cg group [0.33 log_10_(CFU/mL)], while the highest increase was observed for the CaCgCt group [1.43 log_10_(CFU/mL)] (Figure 1A). Mixed ANOVA demonstrated no significant interaction (*p* = 0.616) among the recovery period, group, and gender, but a significant interaction (*p* < 0.001) was found between recovery period and group (Figure 1A). The negative control group demonstrated significant differences (*p* ≤ 0.023) compared with the Ca, Cg, and CaCgCt groups. Significant differences (*p* ≤ 0.033) were also observed between the Cg group and the others, except for groups Ca and CaCgCt groups (*p* ≥ 0.780). At the end of the experiment, two-way MANOVA did not show a significant interaction (*p* = 0.319, Wilks’ Lambda test) between sex and group on the log_10_(CFU/mL) values from both palate and device, but a significant effect (*p* < 0.001, Wilks’ Lambda test) was observed for groups (Figure 1B). For the log_10_(CFU/mL) from palate, a significant difference (*p* = 0.002) was observed between the CgCt and the CaCgCt groups. For the log_10_(CFU/mL) from device, significant differences (*p* ≤ 0.015) were also verified between the CaCt and the other groups, except the Cg and CgCt groups (*p* ≥ 0.063). When the log_10_(CFU/mL) values from palate and device were compared, a significant difference (*p* = 0.024) was observed only for the CaCg group.

For the fungal recovery (SDA plates) from the rats’ palate and devices at the end of the experimental period, two-way MANOVA demonstrated no significant interaction (*p* = 0.086, Wilks’ Lambda test) between group and sex for the log_10_(CFU/mL) values from both palate and device, but a significant effect was observed only for group (*p* = 0.002, Wilks’ Lambda test). Significant differences (*p* ≤ 0.009) were observed for the values from the palate of the CaCgCt group compared with Ct, CaCt, and CgCt (Figure 2A). No significant differences (*p* ≥ 0.119) were observed for the values from the device.

The colonies grown on CHROMAgar *Candida* were used to presumptively identify the species of *Candida* recovered from palate and devices. We also quantified each species of *Candida* based on the color of the colonies and verified a moderate agreement (*p* = 0.578, Interclass Correlation Coefficient, ICC) between the counts of SDA and CHROMAgar *Candida* for palate and an excellent agreement (*p* = 0.960, ICC) for the device [43]. Additionally, the Bland–Altman plot showed agreement (*p* = 0.149) between the CFU/mL values of SDA and CHROMAgar *Candida* for palate, despite a proportional bias (*p* < 0.001) in the difference values (Appendix A). For the CFU/mL recovered from dentures, although the *t*-test showed no agreement (*p* = 0.049) between SDA and CHROMAgar *Candida* and a proportional bias (*p* = 0.002) was verified, the Bland–Altman plot (Appendix A) showed a good agreement between the CFU/mL values, with only some samples above and below the limits of agreement, suggesting absence of systematic discrepancy between the counts from both culture mediums. Therefore, the log_10_(CFU/mL) values from CHROMAgar *Candida* were submitted to statistical inference. Two-way MANOVA did not show significant interaction (*p* = 0.375) between group and sex for the log_10_(CFU/mL) values from both palate and device, but significant effects were observed for gender (*p* = 0.025) and group (*p* = 0.001). The growth of *Candida* spp. from female rats was significantly higher (*p* = 0.007) than that from male rats for colonies recovered from the palate, while no significant difference (*p* = 0.340) between female and male rats was observed for colonies from the device. When the groups were compared, significant differences (*p* ≤ 0.047) were observed only among different species of *Candida* from different groups (Figure 2B,C). The total number of colonies on CHROMAgar *Candida* was quantified and compared with those grown on SDA. Two-way MANOVA did not show significant interaction (*p* = 0.161) between group and sex on the log_10_(CFU/mL) values from CHROMAgar *Candida* and SDA, but a significant effect was verified only for group (*p* < 0.001). Two-way ANOVA demonstrated neither a significant interaction between group and growth media (*p* = 0.960), nor a significant effect for growth media (*p* = 0.447).

### 3.3. Blood Cell Counts

For the blood cell counts (white blood cells, neutrophils, lymphocytes, and red blood cells), two-way MANOVA did not show significant interaction (*p* = 0.184) between group and sex, but significant effects were verified for group (*p* = 0.013) and sex (*p* < 0.001). The blood cell counts from male rats were significantly higher (*p* < 0.001) than those from female rats. When the groups were compared, significant difference (*p* = 0.008) was only observed between Ca and CgCt groups for red blood cells (Figure 3). Interestingly, the mean values of neutrophils were higher for the groups with dual- and triple-species of *Candida* than those with mono-species of *Candida*, even though these differences were not significant (*p* ≥ 0.975).

### 3.4. Macroscopic and Histopathological Analysis

A morphological change in the distribution of the papillae and bleeding in the gingival sulcus region were observed in most animals after removal of the device. However, no sign of inflammation (edema, erythema) was apparent on the palatal mucosae, except some red spots at the anterior papillae in some animals. Similar findings were observed for females (Figure 4) and males (Appendix A). Bleeding scores in the gingival sulcus are shown in Figure 5. While most rats showed moderate and mild bleeding, those infected with the three *Candida* species seemed to exhibit higher levels of bleeding compared with the other groups.

In the histopathological sections of the central palate, both female (Figure 6) and male (Appendix A) rats with the device showed epithelial thinning and loss of the papillae shape between epithelium and connective tissue. Nonetheless, neither fungal presence nor inflammatory infiltrate were observed in any of the groups.

### 3.5. Microbiome Analysis

A subset of samples was subjected to 16S rRNA gene sequencing analysis to reveal changes in the oral microbial community of rats from the Ca group. The swabs from the palate of seven rats (3 males and 4 females) collected at the baseline (pre-inoculation and before cementing the device) and after four weeks of *C. albicans* inoculation and device use (post-inoculation) were sequenced. Alpha diversity analysis, as observed by Shannon’s index (Figure 7A), revealed no significant difference between the pre- and post-inoculation groups. However, beta diversity analysis, as observed by weighted unifrac distances, resulted in significant difference (*p* = 0.002) between the two groups (Figure 7B).

The analysis of the microbial composition at the genus level further confirmed that the microbial profile of the rats changed remarkably after the introduction of the device with *C. albicans* (Figure 8A). After cementing the device on the rats’ palates with *C. albicans* for a period of four weeks, an increase in the relative abundance of genera, such as *Porphyromonas*, *Veillonella*, *Bacteroides*, and *Peptostreptococcus*, all anaerobic bacteria, was observed (Figure 8B). Of these, the relative abundance increased significantly post inoculation for *Porphyromonas* (1.9% to 26.8%) and *Veillonella* (1.5% to 14.3%). Additionally, a decrease in relative abundance of the most abundant genera in the oral cavity, such as *Haemophilus, Rothia, Streptococcus*, and *Acinetobacter*, all aerobic/facultative anaerobe bacteria, was also observed (Figure 8C). Amongst these, the reduction was significant only for *Haemophilus* and *Rothia*, which decreased from 30.9% to 7.8% and from 14.1% to 1.6%, respectively.

## 4. Discussion

In this study, we tried to induce denture stomatitis (DS) in immunocompetent rats to simulate clinical conditions of denture users without any systemic comorbidity, because only the presence of dentures is considered a predisposing factor for DS [44]. With the methodology employed, our study demonstrated neither macroscopic nor microscopic features of palatal inflammation caused by both the acrylic device attached to the palatal mucosa and the presence of *Candida* spp. This absence of palatal inflammation does not corroborate with previous investigations, in which DS was developed in immunosuppressed rats under antibiotic therapy [25,26,27,28,29,30,31,32,38]. Immunosuppression and changes in the resident microbiota, such as those caused by antibiotics, are predisposing factors for the development of opportunistic fungal infections and have been employed successfully in rodent models of candidiasis [45]. However, as these factors are not always found in DS patients, we tried to replace them by a carbohydrate-rich diet to maintain *Candida* spp. in the animals’ oral cavity [21,46]. Although we have recovered *Candida* spp. from the rats’ oral cavity after four weeks with the acrylic device, the absence of palatal inflammation could be also attributed to the pathogenic property of the strains used. An early study demonstrated that a serotype of *C. albicans*, as well as *C. glabrata* and *C. tropicalis*, did not induce DS in rats [34]. Another investigation showed that the SC5314 *C. albicans* strain is eliminated from the oral mucosa of wild-type mice after seven days [47], in agreement with the absence of fungal cells in the histopathological analysis of our study. A more recent study demonstrated that the diversity of *C. albicans* isolates influenced the development of oral infection in immunocompetent mice [48]. It has been also reported that not employing immunosuppression requires high concentration of fungal inoculum [7,31,33]. However, we pelleted our fungal suspensions and inoculated only the fungal pellet on the animals’ palate in order to achieve a high concentration. Therefore, the reference strain used, and the lack of other predisposing factors (immunosuppression and antibiotic therapy), may explain the absence of palatal inflammation during the induction of DS.

We also compared the pathogenesis of DS in male and female rats, since this oral condition is clinically more prevalent in women [2,3] and previous animal studies of DS employed only male rats [25,26,27,28,29,30,31,32,33,34]. However, the lack of inflammatory response in the animals’ palate did not enable such comparison. The higher prevalence of DS in women is attributed to hormonal condition and atrophy of the mucosa in menopause [49]. In our investigation, male rats showed a higher increase in body mass than females during the experimental period, which explains the difference in blood analysis of male and female rats. Another study observed an increase of neutrophils and a reduction of lymphocytes from male rats after four days with acrylic dentures attached to the palatal mucosa [29].

After the experimental period, there was an increase in the total cultivable microbiota from the oral cavity up to 1.43 log_10_(CFU/mL). This result suggests that the acrylic device and the presence of *Candida* spp. increased the rats’ oral microbiota. Another study also observed an increase of *Candida* spp. and bacteria from acrylic device attached to the rat’s palate after 48 h [28]. A recent investigation demonstrated that dietary sucrose increased the bacterial burden and reduced the alpha diversity of oral microbiome from the tongue of immunocompetent mice, while the presence of *C. albicans* also increased the bacterial burden but did not change the alpha diversity [50]. In immunosuppressed mice, sucrose attenuated the tongue lesions caused by *C. albicans* and changed the bacterial community, reducing enterococci and increasing lactobacilli [50]. Lactobacilli have an antagonist effect on *C. albicans* [51], enhance the clearance of *C. albicans* from the oral cavity [52] and protect mice from candidiasis [53]. Although we did not isolate nor identify lactobacilli, we employed a carbohydrate-rich diet with 40% sugar, which may have attenuated the pathogenicity of *C. albicans*.

The sequencing analysis of a subset of samples, however, did identify significant changes in the microbiota of the rat oral cavity after four weeks with the device and *C. albicans*. As observed in Figure 8A–C, a significant increase in the genera associated with disease was detected along with a significant decrease in relative abundance of health-related genera. In particular, the abundance of members of the genus *Porphyromonas*, which is associated with periodontal inflammation as well as periapical infections, increased more than 13-fold, while the abundance of the oral health-related genera *Haemophilus* and *Rothia* decreased 4- and 9-fold, respectively. This indicates an initiation of dysbiosis in the oral cavity, which may lead to development of disease over time. The shift from aerobic/facultative species to anaerobic bacteria may be attributed to the introduction of the device, because dentures reduce the salivary flow and oxygen and provide an anaerobic microenvironment on the palate. However, as a limitation of this study, the samples from the DC group were not evaluated in the microbiome analysis. Such control should be included in future studies to elucidate if the dysbiosis is solely due to the introduction of the device itself. In this study, the rats were kept for a period of four weeks, which may not be sufficient to see the onset of the disease on a histological level. Another study observed mild symptoms post four weeks; however, after eight weeks DS was more severe and visible [31]. Therefore, future studies should evaluate different strains of *Candida* spp., including clinical isolates, and different time points, both shorter and longer periods of device wearing.

Our investigation also evaluated *C. glabrata* and *C. tropicalis* with or without *C. albicans* to investigate their pathogenic potential in developing DS. A clinical study with 82 *Candida*-positive DS patients demonstrated that individuals who harbor only *C. albicans* on dentures were three times as likely to manifest mild inflammation (type I DS), while patients with mixed *Candida* spp. have a five-fold increased risk of the severest inflammation (type III DS) [54]. However, identification of *Candida* spp. was performed only presumptively [54]. Data from another clinical study found high rate of *C. glabrata* in combination with *C. albicans* isolated from type III DS patients, suggesting that such combination may be involved in the severity of DS [9]. However, other studies demonstrated that *C. glabrata* was not able to cause DS in rats nor oropharyngeal candidiasis in mice, but infection was only possible when combined with *C. albicans* [33,35]. Additionally, few studies have utilized *C. tropicalis* on rodent models of oral candidiasis, even though *C. tropicalis* is the most virulent species after *C. albicans* and the most prevalent non-*albicans Candida* species in tropical regions [37,55]. Other studies have demonstrated that a laboratory strain and a clinical isolate of *C. tropicalis* did not induce DS in male rats [34] and only one from three clinical isolates of *C. tropicalis* was able to infect the tongue of female rats [36]. These scarce results may suggest that the pathogenic variability of strains would influence the development of infection caused by *C. tropicalis*, as observed for *C. albicans* [48]. Nonetheless, our investigation could not demonstrate the potential of *C. tropicalis* combined with *C. albicans* in developing DS in rats, due to the lack of palatal inflammation. An in vitro study has demonstrated that only filamentous non-*albicans Candida* species (*C. tropicalis* and *C. dubliniensis*) were capable of adhering to hyphae of *C. albicans* and developing dual-species biofilms, while non-filamentous species (*C. krusei*, *C. parapsilosis*, and *C. lusitaniae*) did not adhere to *C. albicans* [56]. Therefore, future studies are needed to evaluate the pathogenic potential of *C. tropicalis* with *C. albicans* in susceptible hosts.

In conclusion, our investigation demonstrated that an acrylic device attached to the palatal mucosa and the presence of *Candida* spp. did not induce DS in immunocompetent rats, although an increase in disease-related bacterial genera in the oral microbiome was observed upon infection with *C. albicans*. In future investigations, increasing the time in contact with the various *Candida* spp. might help in the induction of DS. Although only the use of dentures is considered a predisposing factor for DS in humans, especially when combined with poor hygiene and continuous denture use (not removing dentures to sleep) [44], other predisposing factors seem to be important for developing DS in rats.

## Figures and Tables

**Figure 1 jof-08-00466-f001:**
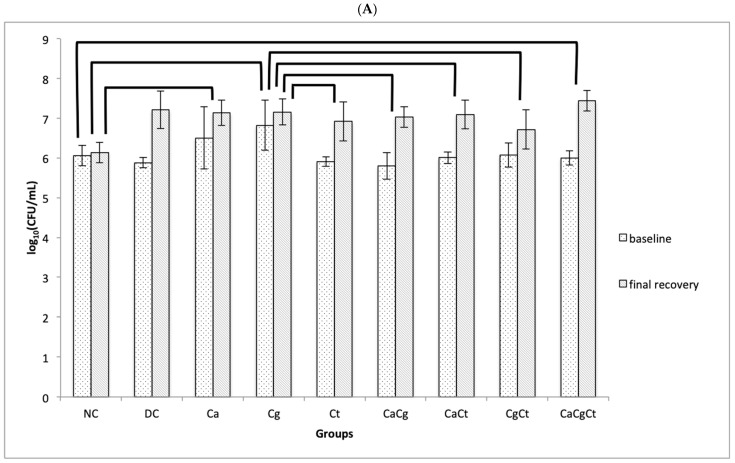
Mean values of log_10_(CFU/mL) of the total cultivable microbiota (BHI) from the rats’ palate and device: (**A**) Initial (baseline) and final recovery from the rats’ palate; (**B**) Final recovery from the palate and the acrylic device. Error bars: standard deviation (*n* = 10, males and females); brackets show significant difference (*p* < 0.05) among the groups. NC: negative control; DC: device control, Ca: *Candida albicans*, Cg: *Candida glabrata*, Ct: *Candida tropicalis*, CaCg: *C. albicans* + *C. glabrata*, CaCt: *C. albicans* + *C. tropicalis*, CgCt: *C. glabrata* + *C. tropicalis*, and CaCgCt: *C. albicans* + *C. glabrata* + *C. tropicalis* groups.

**Figure 2 jof-08-00466-f002:**
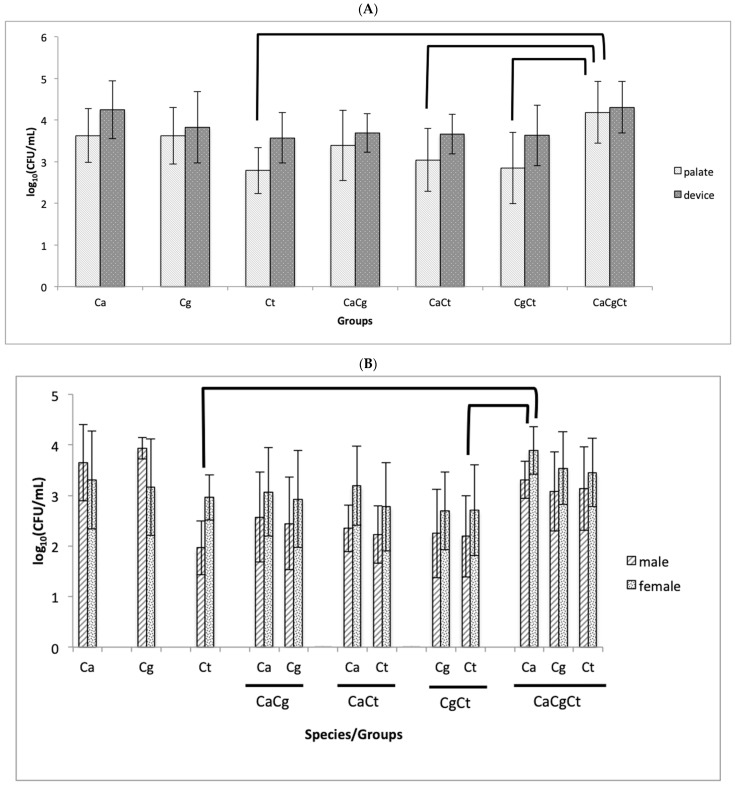
Mean values of log_10_(CFU/mL) of fungi from SDA (**A**) and *Candida* spp. from CHROMAgar *Candida* recovered from the palate (**B**) and the device (**C**) after 4 weeks. Error bars: standard deviation (*n* = 10 for (**A**), male and female, and *n* = 5 for (**B**,**C**)); brackets show significant difference (*p* < 0.05) among the groups (only for palate in **A** e for both sexes in (**B**,**C**). Ca: *Candida albicans*, Cg: *Candida glabrata*, Ct: *Candida tropicalis*, CaCg: *C. albicans* + *C. glabrata*, CaCt: *C. albicans* + *C. tropicalis*, CgCt: *C. glabrata* + *C. tropicalis*, and CaCgCt: *C. albicans* + *C. glabrata* + *C. tropicalis* groups.

**Figure 3 jof-08-00466-f003:**
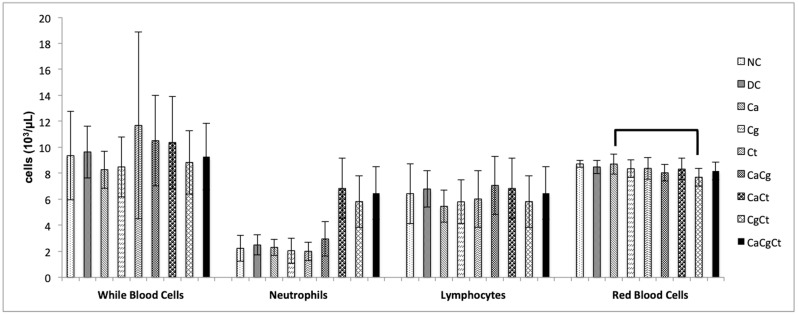
Blood cell counts of male and female rats for each group (*n* = 10). Brackets show significant difference between the groups (*p* < 0.05). NC: negative control; DC: device control, Ca: *Candida albicans*, Cg: *Candida glabrata*, Ct: *Candida tropicalis*, CaCg: *C. albicans* + *C. glabrata*, CaCt: *C. albicans* + *C. tropicalis*, CgCt: *C. glabrata* + *C. tropicalis*, and CaCgCt: *C. albicans* + *C. glabrata* + *C. tropicalis* groups.

**Figure 4 jof-08-00466-f004:**
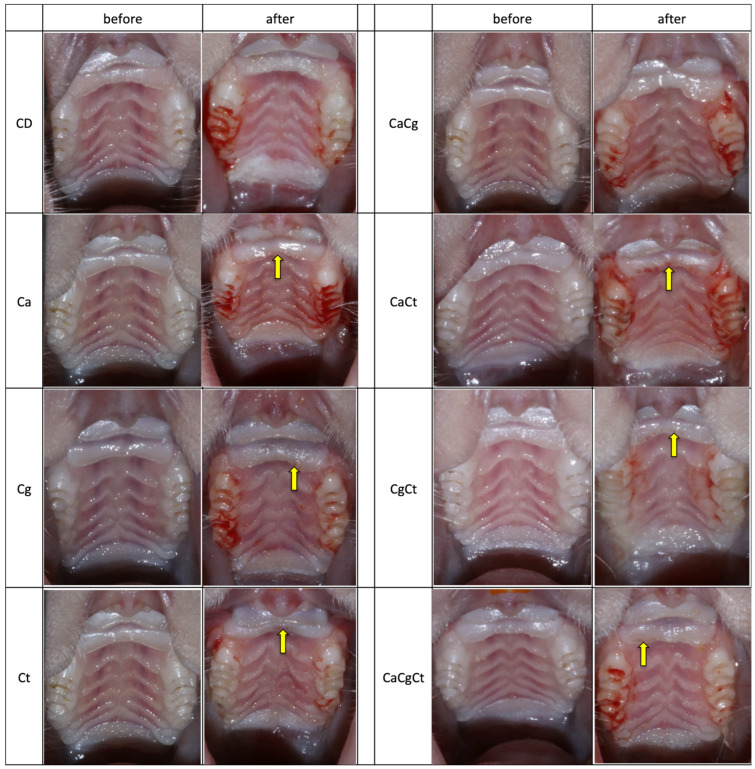
Photographs of the palate of female rats before (day 0) and after (day 42) the 4-week period with the acrylic device. In each group, both photographs (before and after) are from the same animal. The yellow arrows show red spots on the anterior papillae. CD: device control, Ca: *Candida albicans*, Cg: *Candida glabrata*, Ct: *Candida tropicalis*, CaCg: *C. albicans* + *C. glabrata*, CaCt: *C. albicans* + *C. tropicalis*, CgCt: *C. glabrata* + *C. tropicalis*, and CaCgCt: *C. albicans* + *C. glabrata* + *C. tropicalis* groups.

**Figure 5 jof-08-00466-f005:**
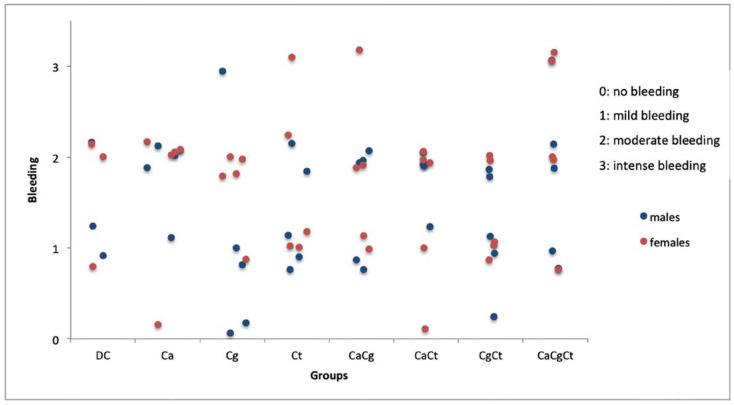
Bleeding of the gingival sulcus region after removal of the acrylic device. DC: device control, Ca: *Candida albicans*, Cg: *Candida glabrata*, Ct: *Candida tropicalis*, CaCg: *C. albicans* + *C. glabrata*, CaCt: *C. albicans* + *C. tropicalis*, CgCt: *C. glabrata* + *C. tropicalis*, and CaCgCt: *C. albicans* + *C. glabrata* + *C. tropicalis* groups.

**Figure 6 jof-08-00466-f006:**
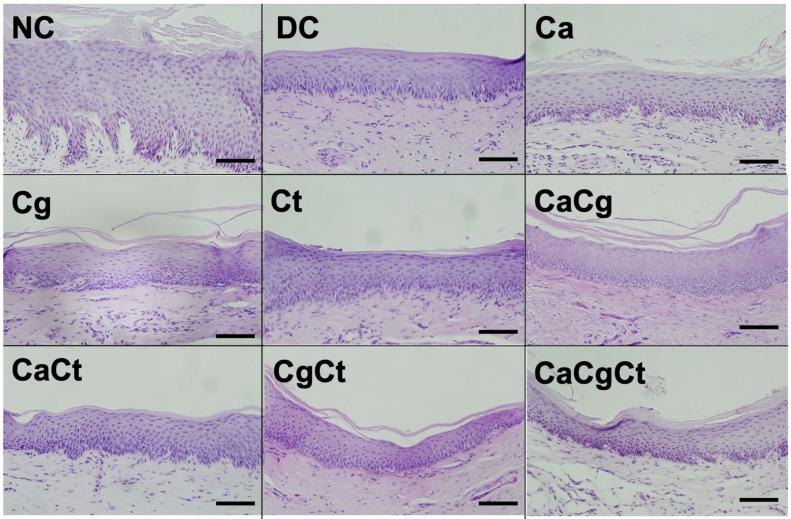
Histopathological sections of the palatal mucosa of female rats after the 4-week period with the acrylic device. NC: negative control. DC: device control, Ca: *C. albicans*, Cg: *C. glabrata*, Ct: *C. tropicalis*, CaCg: *C. albicans + C. glabrata*, CgCt: *C. glabrata + C. tropicalis*, CaCt: *C. albicans + C. tropicalis*, and CaCgCt: *C. albicans + C. glabrata + C. tropicalis*. Magnification: 20×. Scale bar: 100 µm.

**Figure 7 jof-08-00466-f007:**
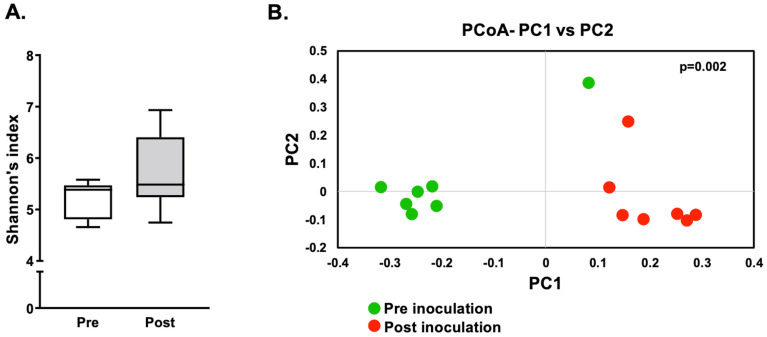
Alpha and Beta diversity: (**A**) Alpha diversity analysis of oral microbial communities of rats as measured by Shannon’s index. Box-plots represent the alpha diversity of species in oral cavity of rats before (pre-white box-plot) and after (post-grey box-plot) cementing the acrylic device with *Candida albicans* on the hard palate. The whiskers represent minimum to maximum values and a line in the box represent the median. (**B**) Beta diversity analysis of oral microbial communities formed in the rat oral cavity as evaluated by weighted unifrac. Principal component analysis (PCoA) is plotted between the samples before (pre) and after (post) cementing the acrylic device with *C. albicans*. The p-value obtained by analysis with 999 permutations in ANOSIM is mentioned within the panel.

**Figure 8 jof-08-00466-f008:**
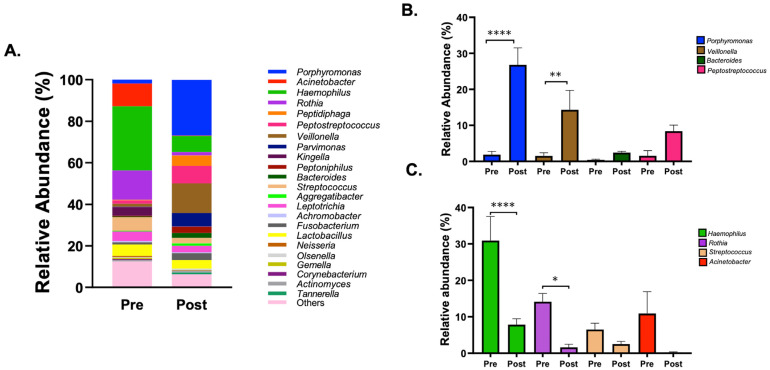
Microbial composition: Genus level microbial composition of the rats’ oral cavity as revealed by 16S rRNA gene sequencing. (**A**) The bar plots represent the relative abundance of genera present in the oral cavity of rats before (Pre) and 4-week after (Post) cementing the acrylic device with *C. albicans*. (**B**) Bar plots representing some of the genera that increased in the relative abundance post cementing the device with *C. albicans* (**C**) Bar plots representing some of the genera that showed decreased relative abundance post cementing the device with *C. albicans*. **** indicates *p* < 0.0001, ** indicates *p* < 0.01 and * indicates significant differences of *p* < 0.05.

## Data Availability

Sequence data will be available upon request.

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
