# Peer review of "A Denture Use Model Associated with Candida spp. in Immunocompetent Male and Female Rats"

_jof, 2022, doi:10.3390/jof8050466_

Round 1

Reviewer 1 Report

The authors characterize a rat denture model in which the animals are fed high sugar diets but no antibiotic or immunosuppressive therapy.  The observations do not identify stomatitis, but do find that Candida appears to alter the prokaryotic oral microbiome.  The group examined the impact of Candida species and rat gender, but differences were not so large.

I have a few questions:

Title: stomatitis was not observed.

Abstract: the gender differences do not seem biologically relevant.

Lines 117-118, the variability seems quite large (10exp5 to 10exp6?  Is this a reporting error?

Line 246, the authors describe 11 deaths (out of how many total animals).

Figure 7 SEM is not particularly illustrative in that organisms or biofilm are not particularly visible.

Author Response

“English language and style   (x) English language and style are fine/minor spell check required”

            Thank you for the considerations and careful revision of our manuscript. The native English-speaking authors RL and BS revised English.

“The authors characterize a rat denture model in which the animals are fed high sugar diets but no antibiotic or immunosuppressive therapy. The observations do not identify stomatitis, but do find that Candida appears to alter the prokaryotic oral microbiome. The group examined the impact of Candida species and rat gender, but differences were not so large.

I have a few questions:

Title: stomatitis was not observed.

            Thank you for the careful revision of our manuscript. We altered the title removing the word “stomatitis”. Now, the title is “A denture use model associated with Candida spp. in immunocompetent male and female rats”.

“Abstract: the gender differences do not seem biologically relevant.”

            We added this information in the abstract.

“Lines 117-118, the variability seems quite large (10exp5 to 10exp6?  Is this a reporting error?”

            The standard deviation of the inoculum concentration for each species was calculated based on the CFU/mL values obtained for each inoculum used in each experiment. These CFU/mL values were not log transformed, so the variability seems to be high in their original data. However, if the values are log transformed, they become low. For example, if the inoculum concentrations of C. albicans in three experiments were 1x106, 2x106, and 3x106 CFU/mL each, the standard deviation would be 1x106 CFU/mL. However, if these values are log transformed, they will be: 6, 6.30, and 6.48 log10(CFU/mL), respectively, which result in a standard deviation of 0.24 log10(CFU/mL). Although we use log transformation only for statistical analysis to meet the assumptions of normality and homoscedasticity, we also added the log values of the inoculum concentrations in the manuscript.

“Line 246, the authors describe 11 deaths (out of how many total animals).”

            We did not include the number of deaths in the total number of animals in the first version of the manuscript. We corrected now the total number of animals, including the number of deaths, because the dead rats were replaced by other animals to keep the n = 5 for each sex for each group. We also excluded the 16 animals used in Scanning Electron Microscopy (SEM, see the question/answer below), and then the total numbers of animals now are 47 males and 44 females.

“Figure 7 SEM is not particularly illustrative in that organisms or biofilm are not particularly visible.”

            We agree with the reviewer, so we removed the SEM analysis from the manuscript.

Reviewer 2 Report

I guess is a nobel point of view  in fungus infections. Experiment is well stablish and a deep analysis of the issue.

Images refect very well experiment

Author Response

“I guess is a nobel point of view in fungus infections. Experiment is well stablish and a deep analysis of the issue.

Images refect very well experiment”

            Thank you so much for the revision and the kind compliments to our investigation.

Reviewer 3 Report

Since, denture stomatitis  is a common infection in denture wearers with unclear etiology, there is a need for this kind of studies. This is well written study, with few limitations, that evaluated the induction of DS using acrylic devices attached to the palate of rats combined with inoculation of Candida spp. Although this results refer to 4 week period of device wearing, my main concern of this study is need to think of inclusion of different time points. I would suggest further studies involving different time points, both shorter and longer since some Candida can induce changes in one week period and also since there was no obvious clinical signs of DS in experimental animals, period of denture wearing should be prolonged. 

Author Response

“English language and style   (x) English language and style are fine/minor spell check required”

            The native English-speaking authors RL and BS revised English.

Since, denture stomatitis  is a common infection in denture wearers with unclear etiology, there is a need for this kind of studies. This is well written study, with few limitations, that evaluated the induction of DS using acrylic devices attached to the palate of rats combined with inoculation of Candida spp. Although this results refer to 4 week period of device wearing, my main concern of this study is need to think of inclusion of different time points. I would suggest further studies involving different time points, both shorter and longer since some Candida can induce changes in one week period and also since there was no obvious clinical signs of DS in experimental animals, period of denture wearing should be prolonged.

            Thank you for the suggestions and careful revision of our manuscript. We agree with the reviewer that future studies should evaluate longer period of device wearing and different strains of Candida. We add this information at the end of the Discussion section of the manuscript. Currently, in another in vitro study we are using a clinical isolate of C. albicans able to grow as hyphae and pseudo-hyphae very easily. Maybe this strain could induce denture stomatitis in a shorter period and is an interesting strain for an in vivo study.

Reviewer 4 Report

see attached file

Author Response

“English language and style   (x) English language and style are fine/minor spell check required”

            The native English-speaking authors RL and BS revised English.

The manuscript by Sakima et al. deals with the attempt to induce denture stomatitis in immunocompetent male and female rats inoculated with different Candida species (single or mixed inocula).

Animal models have been widely used to study oral candidiasis, but the authors chose to use immunocompetent rats to better mimic the status of immunocompetent denture users and focused on the comparison of genders, since this oral condition is clinically more prevalent in women.

The manuscript is overall well written, but some clarifications are needed (see below).

As the authors state, Candida spp. and acrylic device did not induce palatal inflammation macroscopically nor microscopically and the lack of inflammatory response in the animals’ palate did not enable gender comparison. In Discussion section, authors try to give explanations (first, the lack of predisposing factors and, as said in lines 409-410: “the absence of palatal inflammation could be also ascribed to the pathogenic property of the strain used”).

Apart from that, the authors claim (Abstract, lines 24-27): “The microbiome results indicate an increase in inflammatory microbiota and reduction in health-associated microorganisms. Although Candida spp. and acrylic device did not induce DS in immunocompetent rats, the shift in microbiota may precede manifestation of inflammation.” See also lines 446-448 and 453-456 in Discussion section: “The sequencing analysis of a subset of samples, however, did identify significant changes in the microbiota of the rat oral cavity after 4 weeks with the device and C. albicans.” “This indicated an initiation of dysbiosis in the oral cavity which may lead to development of disease over time. In this study, the rats were kept for a period of 4 weeks, which may not be sufficient to see the onset of the disease on a histological level.”.

This claim is not fully justified since microbiome analysis was carried out only in the oral microbial community of rats from the Candida albicans infected group, so that an important control is lacking, i.e. rats with intraoral device and without fungal inoculation.
In the absence of this control it cannot be ruled out that the remarkable change in microbial profile of the rats is due solely to the introduction of the device.

            Thank you for the considerations and careful revision of our manuscript. As mentioned by the reviewer, the microbiome analysis was carried out only in the groups of rats with the intra-oral device inoculated with C. albicans. However, the first microbial sample of the oral cavity was swabbed when the oral impression was taken, before cementing the intra-oral acrylic device and before receiving Candida inoculation (baseline). These first samples were named “Pre samples” in the microbiome analysis, while the samples collected from the same animals after the 4-week period of device wearing were named “Post samples”. Therefore, comparing the microbiome of the pre samples with those of the post samples from the same rats, it was possible to verify the initial dysbiosis in the oral cavity, although we cannot attribute it solely to the introduction of the device or the presence of C. albicans, as very well pointed out by the reviewer. We fully agree with the reviewer and would like to acknowledge that there is a possibility that the changes can be solely due to the introduction of device, because dentures reduce saliva flow and oxygen and create an anaerobic microenvironment on the palate. However, this study was hampered by the lockdown during the beginning of the pandemic of Covid-19 and currently it is difficult to include the denture control samples. We tried to clarify it in the manuscript adding this information in the Materials and Methods (item 2.8) and Discussion.

“It should be noted that the increase in the relative abundance was referred to the genera Porphyromonas, Veillonella, Bacteroides, and Peptostreptococcus, all anaerobic bacteria, and the reduction interested the genera Haemophilus, Rothia, Streptococcus, and Acinetobacter, all aerobic/facultatively anaerobic bacteria. Could it be that the presence of the device (alone) could alter the conditions of the microenvironment and cause this change?

Such a control group should be added to the experiment.”

            As the reviewer well suggested, we added the expressions “all anaerobic bacteria” for Porphyromonas, Veillonella, Bacteroides, and Peptostreptococcus and “aerobic/facultative anaerobic bacteria” for Haemophilus, Rothia, Streptococcus, and Acinetobacter. As mentioned before, dentures reduce the oxygen and salivary flow, creating an anaerobic microenvironment, which may thrive the anaerobic species. We agree with the reviewer that the denture control group should have been added in the microbiome analysis. However, it was not possible due to the reason explained above (lockdown at the beginning of the pandemic) and high cost of sequencing. Moreover, the grant received by EGOM (Capes-PrInt-Unesp, Junior Visiting Professor, process number 88887.468793/2019-00) was for a short-term visit from January to March of 2020, when the in vivo study had not been concluded. So, we selected only the group of rats with C. albicans, males and females, and only the rats whose samples pre and post device wearing were adequate, i.e., samples with acceptable DNA quantity, purity, and integrity. Moreover, one sample showed low quality sequence and could not be used in the sequence data analysis. Therefore, we could only analyze a subset of 3 male and 4 female rats. Therefore, now we reported the absence of the device control group in the microbiome analysis as a limitation of our study in the Discussion.

“Other issues
Materials and Methods section

Paragraph 2.3.
It is not clear if the standardized concentrations indicated in line 118 are (for each yeast strain) the final concentrations present in each well of the 24-well plate (2 mL volume) for biofilm formation on individual devices (indicated in line 124) as well as the final concentrations for oral inoculation (10 mL volume) (indicated in line 130). Please clarify.”

            All fungal suspensions were grown at the mid-log phase to standardize their concentrations at those values (average ± standard deviation) mentioned in the Materials and Methods (section 2.3). These standardized fungal suspensions were used for both, oral inoculations and biofilm formation on the device, and individual fungal growth was used for biofilm formation and oral inoculation. This information was added in the manuscript.

Lines 132-133 and 136-137: it is not clear the amount of suspension pipetted on the animal’s palate and (additionally) through the orifice of the device. Was the pellet resuspended in PBS after centrigugation? Please specify.

            No, the pellet was not resuspended in PBS after centrifugation for oral inoculation. The entire pellet was pipetted on the palate (first inoculation before cementing the acrylic device) or through the orifice of the device (second and third inoculations). Therefore, the amount of suspension varied according to the pellet formed, but volumes were around 50 μL. Moreover, individual fungal inocula (single or mixed) were grown and used for each rat (one pellet for animal). The sentence was rewritten and the information was added in the manuscript.

“Results section

Paragraph 3.1

Figure S1 and its legend are not in agreement with Paragraph 2.3. of Material and Methods section. “Day 0: cementation and first inoculation; day 10: second inoculation; day 13: third inoculation; days 21, 28, and 35: 1st, 2nd and 3rd week after inoculations, respectively; and day 42: euthanasia.” It would seem that first yeast inoculation was made at day 0, second inoculation 10 days later and third inoculation on day 13. If the device was maintained for 4 weeks prior to euthanasia, was it cemented on day 14? In lines 133-138 it was stated: “Immediately after oral inoculation, the device was cemented on the molar maxillary tooth using a self-curing acrylic resin. Additional oral inoculations were performed three times in an interval of three days, and the fungal pellet was introduced into the cavity of the palate through the orifice of the device after it was cemented inside the oral cavity. The devices were kept inside the rats’ oral cavity for 4 weeks.” Times, and even number of inoculations do not agree at all, so that it is not clear how the entire work has been performed. Please clarify!”

            The legend of the Figure S1 was corrected. On day 0, oral impressions were taken, while cementation and first inoculation were performed on day 7 (not day 0), as described in the main text (Materials and Methods, section 2.3).

“Line 246: “During the entire experimental period, 11 animals died (7 males and 4 females).” To which groups did the dead animals belong? If 11 animals died before the end of the 4-week period, it can not be true for all groups the statement in Figure legends (1, 2, 3) “n = 10, males and females” or “n = 10 for A, male and female, and n= 5 for B and C)” or “n = 10”. The same doubt arises for the number of animals indicated by dots in figure 5.”

            The experiments were performed with 5 male and 5 female rats. When animals died, they were subsequently replaced by other animals to maintain the same number of rats in each group, as clarified for the reviewer 1. This information was added in the Manuscript. Therefore, the number of animals in the Figure legends is correct. However, we did not register to which groups the dead animals belonged.

“Paragraph 3.2. Recovery of microorganisms

The presentation of the results, with particular regard to significant differences description, is difficult to follow. As an example, the “significant increase (p < 0.001)” reported in line 249 is referred to each group (i.e. in each group there was a significant difference between baseline and final recovery)? It is not clear. Again, in lines 255-256: “The negative control group demonstrated significant differences (p ≤ 0.023) compared with the Ca, Cg, and CaCgCt groups.” it should be intended for baseline or final recovery or both? Again, it is not clear.”

            The significant increase reported in line 249 refers to all groups, because mixed ANOVA involves “within-subject” factor (recovery period) and “between-subject” factors (gender and group). Therefore, the significant increase in the cultivable microbiota refers to the difference in the recovery period considering all groups and both sexes, because no significant interaction was observed among the three factors. We complemented the information in the sentence in line 249 to clarify it.

            Regarding the significant interaction between groups and recovery periods, when a factorial ANOVA or mixed ANOVA is run on SPSS, the output of the analysis shows only the significant interactions and the significant factors of the analysis. The post-hoc multiple comparisons are shown only for one factor, group in this case, considering both recovery periods and both sexes. Unfortunately, the multiple comparisons considering multiple factors (group and recovery period) are not shown; it is a limitation of SPSS.

“Moreover, brackets intended to show significant difference among the groups in Figures 1A, 2B, and 2C are not clearly understandable.”

            As explained above, SPSS does not show multiple comparisons for more than one factor in its output for mixed ANOVA. In the same way, the output of MANOVA in SPSS does not show the multiple comparisons considering all the dependent variable (CFU/mL from palate and from the device). Unfortunately, it is a limitation of SPSS. Therefore, the brackets in Figures 1A, 2B and 2C refer only to significant differences among the groups, considering both recovery periods (1A) and both sexes (2B and 2C).

“In Figures 4 and S3 the coupled photographs, before and after, are from the same animal?”

            Yes, the photographs (before and after) are from the same animal. This information was added in the legend.

“Figures S4 and S5: While in Figure S5 legend Bar indication is lacking, in Figure S4 Legend a double indication is present (“Scale bar: 100 μM. Scale bars: 100 or 500 μM.”).”

            The indication in the legend of Figure 4 was corrected, while Figures 7 and S5 (SEM analysis) were removed from the manuscript after considering the comment of the reviewer 1.

Round 2

Reviewer 1 Report

The authors addressed questions and suggestions

Reviewer 4 Report

The authors have addressed the raised issue.